# Molecular and Phylogenomic Analysis of a Vancomycin Intermediate Resistance USA300LV Strain in Chile

**DOI:** 10.3390/microorganisms12071284

**Published:** 2024-06-25

**Authors:** Daniela Núñez, Pablo Jiménez, Marcelo Cortez-San Martín, Carolina Cortés, Matías Cárdenas, Sofia Michelson, Tamara Garay, Maggie Vecchiola, Alejandra Céspedes, Jonathan E. Maldonado, Yesseny Vásquez-Martínez

**Affiliations:** 1Molecular Virology and Pathogen Control Laboratory, Departamento de Biología, Facultad de Química y Biología, Universidad de Santiago de Chile (USACH), Santiago 9170022, Chile; daniela.nunez.ac@usach.cl (D.N.); marcelo.cortez@usach.cl (M.C.-S.M.); carolina.cortes.g@usach.cl (C.C.); matias.cardenas.p@usach.cl (M.C.); sofia.michelson@usach.cl (S.M.); 2Laboratorio de Multiómica Vegetal y Bioinformática, Departamento de Biología, Facultad de Química y Biología, Universidad de Santiago de Chile (USACH), Santiago 9170022, Chile; pablo.jimenez.a@usach.cl; 3Escuela de Medicina, Facultad de Ciencias Médicas, Universidad de Santiago de Chile (USACH), Santiago 9170022, Chile; tamara.garay@usach.cl (T.G.); maggievec@gmail.com (M.V.); alejandra.cespedes@redsalud.gov.cl (A.C.); 4Millennium Institute for Integrative Biology (iBio), Departamento de Genética Molecular y Microbiología, Facultad de Ciencias Biológicas, Pontificia Universidad Católica de Chile, Santiago 8380000, Chile

**Keywords:** MRSA, antibiotic resistance, molecular characterization, WGS

## Abstract

Antimicrobial resistance is a major global health problem, and, among Gram-positive bacteria, methicillin-resistant *Staphylococcus aureus* (MRSA) represents a serious threat. MRSA causes a wide range of infections, including bacteremia, which, due to the limited use of β-lactams, is difficult to treat. This study aimed to analyze 51 MRSA isolates collected in 2018 from samples of patients with bacteremia from two hospitals of the Metropolitan Health Service of Santiago, Chile, both in their resistance profile and in the identification of virulence factors. In addition, genomic characterization was carried out by the WGS of an isolate that was shown to be the one of greatest concern (N°. 42) due to its intermediate resistance to vancomycin, multiple virulence factors and being classified as ST8 PVL-positive. In our study, most of the isolates turned out to be multidrug-resistant, but there are still therapeutic options, such as tetracycline, rifampicin, chloramphenicol and vancomycin, which are currently used for MRSA infections; however, 18% were PVL positive, which suggests greater virulence of these isolates. It was determined that isolate N°42 is grouped within the USA300-LV strains (ST8, PVL+, COMER+); however, it has been suggested that, in Chile, a complete displacement of the PVL-negative ST5 clone has not occurred.

## 1. Introduction

Antimicrobial resistance is a major global health concern, and among the Gram-positive bacteria, drug-resistant *Staphylococcus aureus* (*S. aureus*) is a severe threat. *S. aureus* causes a wide range of infections commonly involving the skin, soft tissues, bones, and infections associated with indwelling catheters or prosthetic devices [1]. In addition, *S. aureus* is a leading reason of bacteremia, causing metastatic infections such as infective endocarditis, septic arthritis, and osteomyelitis; moreover, it can result in complications such as septic shock [2]. These issues make *S. aureus* bacteremia particularly challenging to treat, especially if they are methicillin-resistant *S. aureus* (MRSA) due to the limiting use of β-lactams. Methicillin resistance is mediated by the *mecA* gene and acquired by the horizontal transfer of a mobile genetic element designated as staphylococcal cassette chromosome *mec* (SCC*mec*), which confers high variability. The *mecA* gene encodes penicillin-binding protein 2a (PBP2a), an enzyme responsible for crosslinking the peptidoglycans in the bacterial cell wall [3]. PBP2a has a low affinity for β-lactams, resulting in resistance to the entire class of these antibiotics [4].

MRSA has been associated with infections in the community (CA-MRSA) and healthcare settings (HA-MRSA). CA-MRSA bacteremia, including healthcare-associated community-onset, has replaced HA-MRSA bacteremia globally [5]. It has been described that these strains can be distinguished genetically using typing methods due to the presence of different mobile genetic elements (MGE), like SCC*mec* [6]. SCC*mec* are highly variable genomic elements whose sizes range from 20 to 60 kb. However, they all share a three-component structure: (1) a *mec* complex related to methicillin resistance, (2) a *ccr* complex that allows for the excision and integration of the cassette into the chromosome of methicillin-susceptible *S. aureus* (MSSA) strains and (3) three joining regions described as hypervariable regions with non-essential genes such as resistance to other pharmacological families of antimicrobials and resistance to heavy metals [7]. The differences among the *mec* and *ccr* complexes allow for SCC*mec* typing, while the characterization of joining regions gives different subtypes [4].

HA-MRSA strains often contain SCC*mec* type II; in contrast, SCC*mec* type IV is more prevalent in CA-MRSA strains [8]. Other molecular features that distinguish HA-MRSA from CA-MRSA include the presence of the virulence factors acquired by mobile genetic elements (MGEs), such as Panton–Valentine leucocidin (PVL) [9]. The pore-forming exotoxin PVL, encoded by the *lukSF-PV* gene, can facilitate the destruction of leukocytes by forming pores in their membranes, allowing for the influx of ions and small molecules and leading to cell death [10]. The presence of the gene that codes for PVL is a marker of CA-MRSA and is associated with increased disease severity [11].

PVL is not the only toxin produced by MRSA. Different strains produce a range of virulence factors, including toxic shock syndrome toxin-1 (TSST1), exfoliative toxin A (eta), and adhesins like collagen adhesin (Cna), SrdC, and SrdD, that have been found most significantly in MRSA than those among MSSA isolates [12,13]. Those adhesins play an essential role in virulence by establishing a foothold in host tissues and evading the host’s immune system. Adhesins are also crucial in forming biofilms that contribute to persistent infections [14]. Overall, the expression of virulence factors is higher in CA- than in HA-MRSA strains; therefore, CA-MRSA strains tend to be more virulent, as has been described for USA300 strains [15]. In the United States (US), HA-MRSA infections are generally caused by the USA100 or USA200 strains, whereas CA-MRSA infections are commonly associated with the USA300 strain [16].

The USA300 strain (ST8 SCC*mec* IV PVL+) is a very successful clone in its propagation, displacing the previously predominant clone, USA400, in the US [17]. This clone has spread rapidly in countries such as Canada, Switzerland, Belgium, Denmark, Germany, England, Italy, South Korea, Japan, China, and Australia, which has high epidemic potential [18]. Meanwhile, a genetic lineage clone closely related to USA300 was found in Latin America, designated as USA300-LV [19], which is prevalent in Colombia, Venezuela, and Ecuador [20]. Molecular analyses suggest that a common ancestor between USA300 and USA300LV may have emerged in the mid-1970s and that the geographical segregation of the clades occurred in the late 1980s, which coincided with the independent MGE for each clone: the arginine catabolic mobile element (ACME) for USA300 variants and the copper and mercury resistance mobile element (COMER) for USA300-LV variants [21]. Phylogenomic analyses based on WGS among Latin American MRSA strains showed three main clades, a majority multiresistance clade A harboring strains of allelic profile ST5, ST105, and ST1011, a less resistance clade B with ST72, ST88, ST97 and ST8 strains harboring USA300 and USA300-LV, and a minor clade C mainly composed of ST30 strains [21].

In Chile, the PVL-negative Chilean/Cordobes ST5 SCC*mec* I clone has shown high prevalence [22]. However, the emergence of the USA300 clone, detected for the first time in 2008, marked a change in the dynamics of MRSA clones worldwide [23]. The Institute of Public Health of Chile (ISP), responsible for monitoring MRSA in Chile, reported that within the samples collected during the years 2012 and 2013, MRSA PVL+ ST8 isolates were found, corresponding to 11% of the total samples [24]. There has been no update on this clone’s progress in Chile since 2016, highlighting the need for constant and specific surveillance to monitor the circulation of these strains.

Given the lack of use of specific techniques for lineage monitoring, such as WGS, phylogenomic analyses, or SCC*mec* typing with MRSA samples collected in more recent years, a gap is created in the ability to monitor the spread and behavior (phenotypic and molecular characterization) of different MRSA lineages in the country. This study aimed to analyze 51 isolates collected in 2018 from samples of patients with bacteremia from two hospitals in the Metropolitan Health Service, Santiago, Chile, analyzing their antibiotic resistance profile and identifying some virulence factors. Furthermore, given the lack of bioinformatics tools applied in the country, genomic characterization by the WGS of an isolate of concern (ST8 PVL-positive) due to its resistance profile (vancomycin-intermediate resistance) and multiple virulence factors was carried out.

## 2. Materials and Methods

### 2.1. Bacterial Strain (Sampling)

In 2018, 51 positive MRSA isolates were collected from blood culture samples from adult patients with bacteremia at two hospitals in the northern and western areas of the metropolitan region of Santiago, Chile. As a prospective observational study, we included patients older than 18 years who had an episode of *S. aureus* bacteremia. Inclusion and exclusion criteria were as follows. Inclusion criteria: Male and female adult hospitalized patients (>18 years old), *S. aureus* in the bloodstream. Exclusion criteria: Polymicrobial bacteremia; relapsed bacteremia, meaning the resolution of clinical features during treatment and recurrence with phenotypically similar isolates as the original episode. A previous episode during current hospitalization was considered a relapse. Blood cultures were incubated in an automated BacT/ALERT^®^ system for seven days; in case of a positivity alert, they were transferred to solid culture in blood and chocolate agar and incubated at 35 °C for 72 h [25]. Colonies were identified by MALDI-TOF using a VITEK^®^MS and Vitek^®^ 2 system. Quality control was performed with *S. aureus* strains ATCC 25923 and ATCC 43300 [26].

This study was approved by the Ethics Committee of the University of Santiago, Chile (approval number 267/2019), and the Ethics Committee of each hospital (approval number 3917).

### 2.2. Antimicrobial Susceptibility Testing

The Kirby–Bauer disk diffusion method was used to estimate the susceptibility of isolates against eight antimicrobial agents: cefoxitin (30 μg), chloramphenicol (30 μg), ciprofloxacin (5 μg), levofloxacin (5 μg), norfloxacin (10 μg), tetracycline (30 μg), gentamicin (120 μg), and rifampicin (5 mg) according to CLSI [25]. Susceptibility to oxacillin and vancomycin was tested by MIC determination and interpreted according to the CLSI [25] using vancomycin hydrochloride and oxacillin sodium salt. Susceptibility testing was performed according to the clinical and laboratory standard institute CLSI 2020 and CLSI guidelines (M100 Performance standards for antimicrobial susceptibility testing 12 th editions), which entailed choosing different antibiotic families according to Medina et al. 2013 [24]. Quality control was performed with *S. aureus* strains ATCC 25923 and ATCC 43300.

### 2.3. Molecular Characterization

#### 2.3.1. Characterization of Virulence Factors

Isolates were cultured overnight in Müller Hinton broth (MHB), and the cells were harvested by centrifugation at 8000× *g* for 1 min. Bacterial genomic DNA was then extracted using a DNeasy Ultra Clean Microbial Kit. Genomic DNA was used as the template for PCR-based screening assays. All isolates underwent an initial molecular characterization to detect staphylococcal virulence genes by PCR assays. The detection of 7 genes encoding the virulence factor was performed using seven specific primer sets: PVL encoding gene (*lukSF-PV*) and TSST-1 gene (*tst*) according to Víquez-Molina et al. [17]; gamma–hemolysin gene (*hlg*) according to Truong-Bolduc et al. [18]; and collagen adhesin (*cna*), exfoliative toxin A (*eta*), and adhesins (*sdrC*, *sdrD*) according to Carpenter et al. [27]. PCR reactions were set to a final volume of 25 µL, containing 0.5 µL of 10 mM of each primer, 2.5 µL of 10 × 5000 DNA polymerase buffer, 0.5 µL of 5 U/µL Paq 5000^TM^ DNA polymerase, 0.2 µL of 10 mM dNTPs, 18.8 µL of DEPC treated water, and 2 µL of extracted DNA according to manufacture instruction. The amplified PCR products were analyzed by 1% agarose gel electrophoresis. Positive and negative controls used in all experiments belonged to the strain collection of San Juan de Dios Hospital’s Clinical Laboratory.

#### 2.3.2. Multilocus Sequence Typing (MLST)

PVL-positive MRSA isolates were further characterized by MLST sequencing using an internal fragment of 7 housekeeping genes to identify the following allelic profiles: triosephosphate isomerase (*tpi*), phosphate acetyltransferase (*pta*), shikimate dehydrogenase (*aroE*), acetyl-coenzyme A acetyltransferase (*yqiL*), carbamate kinase (*arc)*, guanylate kinase (*gmk*), and glycerol kinase (*glp*) according to Enright et al. [28]. After PCR, the products were purified by Wizard^®^ SV Gel and PCR Clean-Up System (Promega). Purified products were subjected to DNA sequence analysis by Macrogen Inc., Seoul, Republic of Korea.

MLST was supported by the public PubMLST database and the Ridom SpaServer to assign isolates to clonal complexes [29]. Each isolate was grouped according to the corresponding clonal complex according to the sequence type.

### 2.4. WGS and Genome Assembly

Genomic DNA was obtained as previously described in Section 2.3.1. Genomic DNA libraries were prepped using the NexteraXT DNA Sample Preparation kit and sequenced in a configuration of 150 bp paired-end reads using the Nextseq 500 System (Illumina Inc., San Diego, CA, USA).

The raw reads were trimmed using Trimmomatic v0.39 [30] at its ends if their quality was below a Phred score of 30, and de novo assemblies were performed using Velvet v1.2.10 [31]. Quality improvement was represented with the BoxPlotR tool [32]. According to statistical metrics N50 and the number of contigs, the most contiguous assembled genome was selected to perform genome and functional annotation with Prodigal v2.6.3 [33], Prokka v1.13 [34] and RAST v2.0 [35,36]. The completeness of the assembled genome was evaluated with BUSCO v5.4.3 [37]. Genome representation was obtained with GenoVi v0.2.16 [38].

The genome sequence of the isolate n°42 was deposited in the NCBI database under the Bioproject ID PRJNA1073500.

For SCC*mec* typification, the tool SCC*mec*Finder v1.2.1 [39], a 90% threshold for ID, 60% minimum length and both the referenced and extended database were evaluated. Mobile elements COMER and ACME were identified following the methodology described by Arias et al. (2017) [21]. Through BlastN [40,41], positive hits were selected if they had an e-value lower than 1 × 10^−100^, at least 90% of identity and a coverage higher than the 80% of the CDSs in the reference sequences for COMER (CP007672.1 region: 53520–77705) and ACME (Accession: CP000255.1 region: 63100–88681). Representation of genomic elements found in the assembled genome was visualized with UGENE v46.0 [42].

For detecting more antibiotic resistance determinants and virulence factors in the assembled genome, the tool VirulenceFinder v2.0 [43,44] was used. Parameters were set at a 90% threshold for ID, 60% of minimum length, and *S. aureus* was selected as a species. Annotated genes found with Prokka and Rast were also considered.

### 2.5. Phylogenomic Analysis

Sequenced MRSA genomes and non-assembled MRSA reads of Chilean human hosts were collected from the NCBI database for the phylogenomic reconstruction (see Appendix A). Additionally, thirty-four USA300-LV ST8 Latin American genomes available from the NCBI database were included in the reconstruction, NCBI accession codes are listed in Appendix A. Reference *S. aureus* strains NCTC8325, HA-MRSA N315 and CA-MRSA CA12 were considered, and the reference strain *Macrococcus caseolyticus* FDAARGOS 868 was included as an outgroup.

Non-sequenced genomes found in the NCBI database were processed as described in the previous section. Assembled genomes with completeness below 90% were not considered for the phylogenomic reconstruction. MLST analysis was carried out with the public PubMLST database for all the assembled genomes.

A core genome was calculated with Roary v3.13.0 [45]. Sequences of the core genome were aligned with MAFFT v7.511 [46] and the alignment was processed in MEGA v10.2.6 [47] to reconstruct the phylogeny with a maximum likelihood method and the Tamura–Nei substitution model with a 200 bootstrap resampling for the first phylogeny reconstructed setting a site coverage cutoff of 99% due to the high amount of informative sites. and a 1000 bootstrap resampling for the second phylogeny. 

## 3. Results

### 3.1. Antimicrobial Resistance

All MRSA isolates (*n* = 51) underwent a minimum inhibitory concentration test (MIC) against nine antibiotics to test antimicrobial resistance. It was found that 94.1% (48/51) were resistant to oxacillin, while three were susceptible, considered as oxacillin-susceptible MRSA (OS-MRSA). Therefore, following the Clinical and Laboratory Standards Institute (CLSI) protocol, these isolates were subjected to a complementary test for susceptibility to cefoxitin (30 g) by the Kirby-Bauer method, which confirmed the same results as with oxacillin. However, interestingly, all 51 MRSA isolates were *mecA* positive. Concerning vancomycin, most of the isolates evaluated showed sensitivity (98%), and just one (isolated n°42) showed intermediate sensitivity.

Furthermore, the isolates were susceptible to tetracycline, chloramphenicol, and rifampin; half of the isolates were susceptible to gentamicin. The isolates mainly showed resistance to norfloxacin, levofloxacin, and ciprofloxacin (Table 1). The results show that 23 isolates are resistant to one or two pharmacological families of antimicrobials, and 27 are multiresistant (resistance to at least one antimicrobial in three or more families). Just one isolate was sensitive to all antibiotics tested.

### 3.2. Characterization of Virulence Factors in MRSA Isolates

Molecular characterization of some virulence factors in the MRSA strains was carried out using the primers shown in Appendix A. It was observed that the predominant virulence factors in the isolates were the genes codified for adhesins: *sdrD* gene and *sdrC* gene. The *pvl* gene was found in 18% of the isolates. On the other hand, only 4% of the isolates have the *cna* gene, and no positive strains were found to have *tsst* and *etA* genes (Figure 1).

### 3.3. Multilocus Sequence Typing (MLST) of PVL-Positive Isolates

The allelic profile for the PVL-positive isolates was obtained, and the sequence types (STs) and clonal complexes (CCs) were determined using PubMLST (https://pubmlst.org/saureus/, accessed on 10 May 2023). In the nine PVL-positive isolates, it was found that the predominant STs were ST5 and ST8. Two clonal complexes were obtained: CC5 and CC8. Table 2 summarizes the antimicrobial resistance profile and the virulence factors in the nine PVL-positive isolates.

Among the nine PVL positive isolates, isolate n°42 was considered a sample of interest because it was the only vancomycin-intermediate isolate with an MLST profile ST8 CC8, including the multiresistance and virulence factors all isolates also presented. Interestingly, the above data suggest that isolate n°42 is related to USA300-LV.

### 3.4. Sequencing and Genome Assembly of Isolate n°42

Due to the importance of understanding and to update the genetic characteristics of MRSA PVL-positive isolates, such as CA-MRSA or USA300/USA300-LV in Chile, we performed the genomic characterization and established their phylogenomic relationship with the USA300-LV clone and others by whole genome sequencing the isolate n°42.

Table 3 shows the quality improvement after raw sequence trimming. Filtered reads improved by 1.2 and 1.4 points in the Phred 30 scale for forward and reverse reads, respectively. Both forward and reverse sequences showed a decrease of 22.6% in the total amount of reads, corresponding to 1.8 million reads discarded and leaving 6.3 million reads available for assembly. A graphic representation of quality improvement is shown in Appendix A.

Using the software Velvet v1.2.10 and the filtered reads, an assembled genome of 2.8 Mb with an average coverage of 290X was obtained. The genome features of this new genome are summarized in Table 4. A range of k-mers sizes was tested, and a size of 111 bp gave the maximum N50 (453,788 bp) and a genome with the minimum number of contigs (49) and L50 (3). The completeness of the genome was evaluated for the first four taxa classes, and the genome showed the completeness of over 99% for the BUSCO groups searched. Through the functional annotation of 2574 protein-coding genes, 12 rRNA and 62 tRNA were found. The genome GC content was 32%, like the other *S. aureus* genomes reported. Figure 2 shows the circular representation of the assembled genome, adding functional annotations from the Cluster of Orthologues Groups (COGs). COG categories are listed in Appendix A.

### 3.5. SCCmec Typification, Virulence Factors and Antimicrobial Resistance Profile of Isolate n°42

By the complete cassette prediction method over the isolate n°42, a region was found within the first 19 kb with 93.7% identity to SCC*mec* IVc 2B from *S. aureus* strain 2314 (GenBank #AY271717.1). By component search method, all the searched components have an identity above 99%. A type 2 *ccr* complex was identified with the allotypes *ccrA2* and *ccrB2*, and a class B *mec* complex was described according to the *mecA* gene, including the truncated *mec*R1 gene and the IS1272 insertion sequence. The typification of the complexes mentioned above placed the SCC*mec* as type IV while, as the gene for an abi-c abortive phage resistance protein was found, we can place this SCC*mec* cassette as a subtype c.

The search for the ACME/COMER components showed that the isolate n°42 possesses the COMER element downstream of the SCC*mec* between 19 and 43 kb, as shown in Figure 3. Each component described in the reference strain *S. aureus* CA12 was found in isolate n°42 with an identity of over 90%. The genes for copper metabolism (*copB*, *mco* and *copL*) and mercury metabolism (*merA*, *merB* and *merR*) were found in the 3′ region, while various proteins and hypothetical unidentified proteins were found in the 5′ region (see Appendix A for alignment results). The search for the components of the ACME element showed that no gene was found in the isolate n°42 except for the *copB* and *copL* genes shared between ACME and COMER.

On the isolate n°42, genes coding for virulence factors such as gamma hemolysin, leucocidins, staphylococcal enterotoxins and exoenzymes such as serine protease and aureolysin (Table 5) were also found. All these elements were identified with an identity percentage greater than 99%. The functional annotation carried out by the RAST platform identified 22 genes which were classified within the adhesin subsystem, coding mainly for surface proteins of staphylococci, serine–aspartate repeat proteins for the binding of extracellular matrix proteins and proteins associated with coagulation such as fibronectin- and fibrinogen-binding proteins and clumping factor A. Of the seven virulence factors evaluated by PCR, within the genome of isolate n°42, only the genes corresponding to the PVL toxin and the adhesins SdrC and SdrD were found.

The search for genes coding for antimicrobial resistance besides methicillin showed the presence of response genes to several antibiotics, as seen in Table 6, including bacitracin, quinolones, fosfomycin, tetracycline and vancomycin resistance (*vraR* and *vraS*). Some genes associated with response to multiple drugs, such as *mdtG*, *mdtH*, *emrY*, *emrK*, *sepA* and *marA*, were associated with resistance through efflux pump complexes. Response genes to antiseptics such as acraflavin (*acr*) were found.

### 3.6. Phylogomics of Isolate n°42

To discover the genetic diversity of Chilean genome MRSA strains available from NCBI and establish the relationship between the isolate n°42 and the Chilean strains, a phylogenomic tree that included 143 *S. aureus* genomes was constructed (see Appendix A). The first selection criterion considered only MRSA genomes, which would have left only 83 genomes for the phylogenomic reconstruction. A second criterion was the addition of another 59 *S. aureus* genomes from the NCBI database to dissect the population structure of all regional genetic lineages (see Methods).

The phylogenomic analysis revealed that the core genomes were associated according to their MLST profiles, and, as reported by Arias et al. [21], three major clades, A, B, and C, were found (Figure 4). Clade A, which included *S. aureus* N315 (an HA-MRSA reference strain), was the clade that grouped the largest number of isolates, corresponding mainly to ST5 and ST105 profiles. Clade B, which included isolate n°42 and *S. aureus* NCTC8325 ST8 CC8, harbored ST8, ST239 and ST923 profiles. Finally, clade C grouped a low amount of ST30 profiles. Outside of the three major clades, genomes grouped into minor clades corresponding to clonal complexes CC1, CC5, CC8, CC15, CC30, CC45, CC97, and CC121 indicated genotypic diversity.

Since Arias [21] reported that the USA300-LV lineage was part of the ST8 CC8 clade B, we performed a second phylogenomic reconstruction among the 22 Chilean genomes found in clade B, with 34 Latin American ST8 genomes (recovered from the Arias study, see Appendix A) reported as South American USA300-LV and North American USA300 strains. The phylogenomic tree (Figure 5) showed that isolate n°42 was mainly grouped among the USA300-LV Ecuadorian and Colombian strains rather than other USA-300 strains like FPR3757 (a USA300 reference strain).

## 4. Discussion

Bacteremia is arguably the most important infection caused by *S. aureus*, associated with a mortality rate of 14–45%, which increases to 60% when the pathogen is resistant to methicillin [48]. The prevalence of bloodstream infection by MRSA in Latin America appears to be heterogeneous, ranging from 6% in Central America to 80% in some South American countries [49]. The rising prevalence of methicillin resistance and the higher associated mortality compel us to understand the characteristics of MRSA isolates better. Currently, MRSA strains are classified as hospital-acquired and community acquisition [50], presenting molecular characteristics that allow them to be differentiated. Therefore, in this study, the phenotypic, molecular, and phylogenic characterization of MRSA strains was carried out.

The results of 51 Chilean MRSA isolates dated in 2018 show that 94.1% of the isolates (48/51) were resistant to oxacillin, and three were susceptible; however, 100% were *mec*A-positive, confirming the initial identification of MRSA in the laboratories of the respective hospitals. Positive *mec*A strains sensitive to oxacillin (OS-MRSA) have been previously described worldwide [51] and are sensitive to cefoxitin, as in our study [52]. Because OS-MRSA shows low-level β-lactam resistance, patients infected with this MRSA can be treated with β-lactam antibiotics, which could cause the emergence of high-level β-lactam-resistant MRSA. As the clinical diagnosis is almost always performed by phenotypic techniques and not by amplifying the *mec*A gene, it could be erroneously identified as methicillin-sensitive S. aureus (MSSA). Therefore, since MSSA strains are handled differently from MRSA strains, using as, principal antibiotics, β-lactams, it is crucial to investigate and report the occurrence of OS-MRSA to allow for the correct decision-making in treatment, as incorrect treatment could contribute to the development of greater antimicrobial resistance. To our knowledge, this is the first report of an isolated OS-MRSA in Chile.

Despite their resistance to oxacillin and possibly to all currently available β-lactam antimicrobial agents, it is notable that among the Chilean isolates, high levels of susceptibility to chloramphenicol, rifampin, and vancomycin were found. However, interestingly, tetracycline shows 100% antimicrobial efficacy, suggesting that these commonly used antibiotics are an option for treating infections. On the other hand, the isolates showed high levels of resistance to fluoroquinolones, including norfloxacin, levofloxacin, and ciprofloxacin, like a report by Li et al. in 2019 [53], in which MRSA isolates showed significantly higher resistance to levofloxacin than MSSA.

Regarding vancomycin susceptibility, almost all isolates showed sensitivity, suggesting that it remains a therapeutic option to treat MRSA infection. However, one isolate (n°42) showed intermediate susceptibility, which interests us. The prevalence of vancomycin-resistant S. aureus (VRSA) in Europe is 1.1% among 179 isolates, and in America, it is 3.6% among 140 isolates, and the prevalence of vancomycin-intermediate *S. aureus* (VISA) is 1.8% and 1% in Europe and America, respectively [54]. According to the last Public Health Institute report, sensitivity to vancomycin in Chile has been maintained, which showed that 1526 isolates (100%) collected during 2014–2016 were sensitive to this antibiotic [55], suggesting appropriate treatment management. However, this VISA isolate n°42 strongly suggests that the surveillance of vancomycin resistance needs to be maintained.

Concerningly, 27 isolates were considered multidrug-resistant (MDR), defined as non-susceptibility to at least one agent in three or more antimicrobial categories [56], and 23 isolates were resistant to at least one antibiotic. CA-MRSA strains are often reported as MDR and can present resistance to fluoroquinolones, macrolides, aminoglycosides, tetracyclines, and rifampin [57]. This can make treatment more difficult and underscores the need for appropriate antibiotic stewardship practices to preserve the effectiveness of available drugs. Overall, these results highlight the importance of maintaining the ongoing surveillance of antimicrobial resistance patterns to guide appropriate treatment and antibiotic stewardship.

The multi-resistance of the isolates is aggravated by the presence of virulence factors that benefit the infection process. Virulence factors associated with adherence and tissue invasion, such as adhesins, also promote MRSA infection. This study found that the primary virulence factor genes correspond to factors such as adhesins (88% *sdrD*, 41% *sdrC*). These factors have also been reported to be the most common among MRSA isolates in studies of Korean and Mexican strains, with values of 76.2 and 94.5%% for *sdrD* and 92.3 and 89%% for *sdrC*, respectively [58,59].

On the other hand, toxins or proteases secreted by *S. aureus*, like PVL, are essential components in the ability to cause disease. PVL causes leukocyte destruction and tissue necrosis and is present in less than 5% of strains [60]. In our study, 18% were PVL-positive, which suggests greater virulence of these isolates. Although the prevalence of PVL in bacteremia isolates has been reported to be lower than that in soft tissue and skin infections, the presence of PVL has been linked to an increased risk of complications, including infective endocarditis, osteomyelitis, and necrotizing pneumonia [11]. It can be suggested that due to the presence of these virulence factors and the previously mentioned antibiotic resistance such as oxacillin, levofloxacin, norfloxacin, ciprofloxacin, these isolates can cause severe infection, such as bacteremia; however, host factors and the immune response also play a role. In Chile, there are few previous reports on the molecular characterization of virulence genes in MRSA isolates, so the results obtained in this study contribute to the molecular characterization of MRSA virulence factors in this country.

The emergence of PVL-positive MRSA isolates has been reported worldwide. Previous studies revealed a strong association between the presence of PVL genes and CA-MRSA strains, and HA-MRSA strains that carry PVL genes have been reported in various regions in Europe and Asia [61,62]. According to MLST, we report that PVL-positive MRSA isolates correspond to sequence type 8 (ST8) and ST5. USA300 is a ST8-PVL clone characterized by present staphylococcal chromosomal cassette *mec* (SCC*mec*) type IV, ACME, and is a predominant strain in North America [58,63]. In 2005, it was reported that a variant of USA300 lacking ACME was identified in northern South America with clinical and epidemiological characteristics similar to those of the North American clone, suggesting the exportation of USA300 to South America caused epidemics [20]. Between 2006 and 2008, it was reported that this USA300 variant, USA300LV, could ultimately replace previously prevalent hospital-associated clones in Colombia and Ecuador and became the most predominant [20]. Otherwise, isolate ST5 belongs to the Chilean/Cordobes clone that typically carries *SCCmec* I. This clone has been reported widely in several countries in Latin America, as well as Argentina, Brazil, Colombia, Chile, Peru, and Venezuela [59]. USA300 was described in Chile for the first time in 2008 [64] and again in 2017 [65]; however, this clone has not been tracked until this study. The presence of both clones in our isolates suggests the need for a discussion about the ability of ST8 (USA300) to replace ST5 (Chilean/Cordobes), as happened in Colombia and Ecuador [20]. The background suggested the importance of performing a more in-depth molecular characterization, so isolate n°42 was chosen for WGS characterization and to analyze if there are differences from those reported in other South American countries.

Whole genome sequencing has become one of the most recent methods to assist in MRSA’s global and local epidemiological monitoring. The preservation and availability of genomes are crucial in monitoring retrospective and prospective studies. Through WGS and phylogenomic reconstruction, it was possible to compare clinical isolates grouped in clades among USA300-LV strains and identify the isolates responsible for the CA-MRSA outbreak in France 2007 [66]. Using the above-mentioned methodologies, it was determined that isolate n°42 is grouped among USA300-LV strains.

Latin American MRSA strains are grouped into three main clades (A ST5, B ST8 and C ST30) [21]. The similar distribution obtained with the Chilean strains corroborates the prevalence of the most frequent lineages already described within the South American continent, these being the pediatric clone (ST5 SCC*mec* IV), the Chilean/Cordoban clone (ST5 SCC*mec* I), the New York/Japanese clone (ST5 SCC*mec* II), the Brazilian clone (ST239 CC8 SCC*mec* III) and the Oceania-Southwest Pacific clone (ST30 CC30 SCC*mec* IV) [67]. The main abundance of strains for clade A shows the prevalence of the predominant lineages in Chile. However, the abundance of isolates grouped within clade B, where isolate n°42 was found along with isolates collected from 2009–2014, shows the advance of ST8 strains where USA300 and USA300-LV are found [24]. The predominance of these strains would be associated with advantages that SCC*mec* IV provides, such as easier transmission and faster replication due to a smaller SCC*mec*, a lower number of antibiotic resistance genes that represents an advantage for the cost of biological effectiveness and a higher growth rate that allows for more successful colonization against other bacteria [67].

The Chilean ISP CA-MRSA surveillance program considers only the PCR detection of the *mecA* and *pvl* genes, Spa protein typing and MLST, and does not consider SCC*mec* typing or WGS-based analyses [68]. Due to the high plasticity of SCC*mec* components, typing requires WGS and bioinformatic tools since typing based exclusively on PCR can increase the number of non-typeable elements [39]. The rapid expansion of CA-MRSA strains has led to SCC*mec* IVc 2B being the most frequently identified SCC*mec*, and the exclusivity of a class B *mec* complex and a type 2 *ccr* complex allows SCC*mec* typing to identify CA-MRSA strains [4].

The lack of deep sequencing techniques and the exclusive search for the ACME element during the early 2000s led to a late identification of USA300-LV [66]. In Chile, MRSA isolates corresponding to USA300-LV were characterized as ACME-negative strains [24]. The COMER element found downstream of SCC*mec* in isolate n°42 has been described to be present only in USA300-LV, unlike the ACME element, which is widely distributed among MRSA ST8 isolates [69]. Copper is an essential element since it acts as a cofactor within catabolic reactions and facilitates the transfer of electrons for many bacteria. However, its excess can increase the formation of radicals, increasing oxidative damage and potentially causing the death of the bacteria [69]. To prevent that, all *S. aureus* harbor the *copAZ* operon that allows for the export of excess copper; however, the *copBL* and *copBmco* operons found in the COMER element of isolate n°42 have been reported only in some invasive strains such as MRSA252 and ATCC12600 [69]. The *merABR* operon, which mediates the resistance of *S. aureus* to toxic mercury compounds, was found in isolate n°42; the operon allows for resistance to inorganic mercury through the enzyme mercury reductase encoded by the *merA* gene and resistance to organic mercury compounds through the enzyme alkylmercury lyase encoded by the *merB* gene, respectively [69]. Despite the advantages in resistance provided by the COMER element, the absence of the ACME element could imply disadvantages associated with pathogenicity. To date, no studies indicate a direct relationship between the formation of biofilm in USA300-LV associated with the COMER element. Unlike USA300, it has been described that the *speG* gene, present in the ACME element at high concentrations of polyamines, positively regulates genes required for biofilm formation, and comparative studies indicate that strains with the ACME element show more capacity for biofilm formation [69]. Further studies are needed to establish a relation among COMER elements and biofilm formation in CA-MRSA.

According to the virulence factors described in isolate n°42, adhesins and toxins were found, as was previously reported; among them were those identified by PCR, *pvl* genes, and adhesins as *sdrC*, *sdrD* as expected. Related to resistance genes in isolate n°42, four bacitracin resistance genes were found. Bacitracin resistance is prevalent within CA-MRSA strains [70]. For USA300, it has been reported that resistance to bacitracin is accompanied by resistance to neomycin [71]; for isolate n°42, no *neo* gene was found within the genome, only the *bceAB* and *bceRS* genes for the efflux pumps of bacitracin [72].

The *norB* gene, found in isolate n°42, encoding multidrug efflux pumps, is distributed within various strains of *S. aureus,* and for those that present it, they show resistance to a variety of antibiotics, including norfloxacin, and ciprofloxacin [73], which corresponds to the high percentage of resistance found for these antibiotics in the phenotypic characterization.

For MRSA strains, the *tetKM* genotype has been reported as one of the most commonly found and the one that presents the highest levels of resistance through associated protection proteins for ribosomes against antibiotics such as tetracycline, doxycycline or minocycline; however, isolate n°42 only has a *tetAR* gene encoding efflux pumps [74]. Moreover, the RAST platform classification indicates that the genome of this isolate is sensitive to tetracycline, which agrees with the observed phenotypic profile.

The *fosB* gene found in isolate n°42 is considered predominant over others, such as *fosA* and *fosC*, within MRSA strains and other pathogens for resistance to fosfomycin [73].

As mentioned before, vancomycin is one of the antibiotics of choice for treating MRSA infections; however, resistance to this antimicrobial was discovered in enterococci in the 1980s and was associated with the *vanA* operon and in the early 2000s, strains of *S. aureus* resistant to vancomycin VRSA were first reported [55]. From reports of strains with intermediate resistance VISA, heterogeneous h/VISA subsets were described with low resistance to vancomycin and a phenotype that considers increased thickening of the cell wall that would prevent the passage of the antimicrobial. These strains present a resistance mechanism not associated with *van* genes and are not fully described [74,75]. To date, no cases of VRSA have been reported in Chile. In 2015, the first case of a h/VISA strain in the country was reported [55].

For isolate n°42, no *vanA* or *vanB* genes were found in the genome, indicating a resistance phenotype not associated with those genes; instead, the *vraR* and *vraS* genes found in isolate n°42 are described as part of a proposed mechanism for the development of resistance to vancomycin, in which, when faced with vancomycin, the *vraRS* genes induce the expression of *vraCP*, which promotes the expression of genes related to cell wall synthesis such as *glyS*, *sgtB*, *ddl* and *alr2* and hydrolysis genes such as *sceD*, *lytM* and *isaA* [75]. VISA and h/VISA strains are associated with different genetic lineages, including CC5, CC8, CC30 and CC45; both are widely adapted to hospital environments and have a much higher prevalence than VRSA strains. Considering that the things associated with VRSA are pretty scarce, the VISA and h/VISA strains must be addressed as a significant problem, considering also that the h/VISA strains present a rapid reversibility to vancomycin-susceptible strains, VSSA, if the selective pressures caused are reduced by the antibiotic [76].

## 5. Conclusions

In conclusion, despite the high resistance and the fact that most of the isolates are considered MDR, there are still some therapeutic options, such as tetracycline, rifampin, chloramphenicol and vancomycin, which are currently used for MRSA infections. However, these results highlight the importance of maintaining ongoing surveillance of both antimicrobial resistance and virulence factors in MRSA isolates to guide appropriate treatment strategies and prevent the emergence and spread of antibiotic-resistant and virulent strains. Moreover, as could be observed with isolate n°42, the presence of USA300-LV clones (ST8, PVL+, COMER+) can be suggested in Chile as in the other Latin America countries; however, a complete displacement of the PVL-negative ST5 clone has not occurred. Many factors may be contributing to this, such as environmental and genetic factors and the selective pressure caused using antibiotics, so the important thing is to continue monitoring the molecular characteristics and antimicrobial profile of *S. aureus.* The results presented here allow us to demonstrate the phenotypic and genetic characteristics of MRSA strains circulating during 2018 in our town, being relevant to strengthening the epidemiological surveillance systems of our country and region. The study will guide the choice of treatments and design preventive control strategies against the spread of MRSA infection. An important limitation of our study was the participation of only two hospitals in Santiago, so the sample size was small and not representative of the entire country. However, the hospitals participating in this study serve a significant portion of the city’s population. On the other hand, we did not have the support to perform WGS on the entire collection, which could have provided a deeper knowledge of the characteristics of the circulating MRSA strains; nonetheless, relevant information was obtained from the strain studied through WGS that gives support to the conclusions of this study.

## Figures and Tables

**Figure 1 microorganisms-12-01284-f001:**
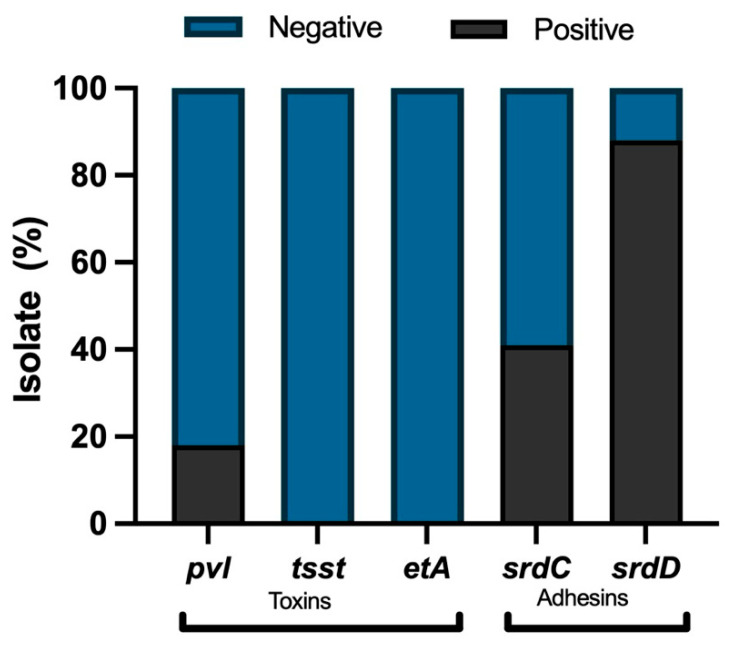
Virulence factor genes in MRSA isolates (*n* = 51): presence (black) and absence (light gray).

**Figure 2 microorganisms-12-01284-f002:**
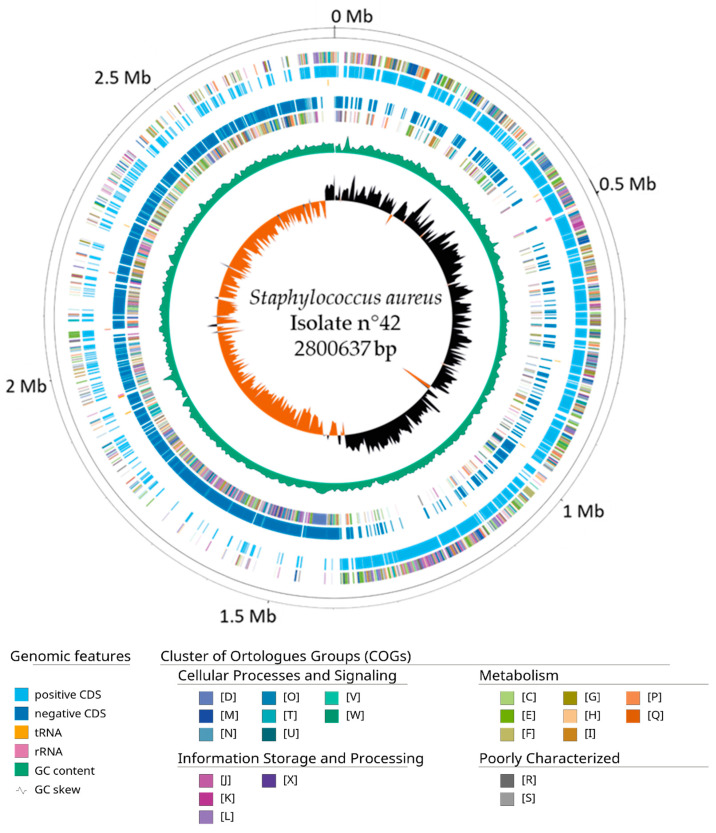
Circular map of the assembled genome of isolate n°42. From outer to inside. Positive COGs, positive CDS (cyan), RNA genes (tRNAs, yellow; rRNA, pink), Negative CDS (blue), Negative COGs, GC content (green) and GC skew. Genome scaffolds were artificially sorted against the *Staphylococcus aureus* CA12 strain reference genome for this visualization to evaluate a correct GC skew distribution over the sequences.

**Figure 3 microorganisms-12-01284-f003:**
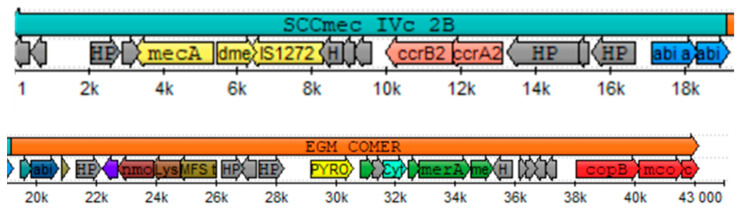
Representation of the Staphylococcal cassette chromosome SCC*mec* IVc 2B and copper and mercury resistance (COMER) mobile element typified in isolate n°42. Color assignment for CDS SCC*mec* components: yellow = complex *mec* genes, pink = complex *ccr* genes, blue = subtype determinants in region J1. Color assignment for element CDS COMER components: red = copper metabolism, green mercury metabolism. Gray = hypothetical proteins.

**Figure 4 microorganisms-12-01284-f004:**
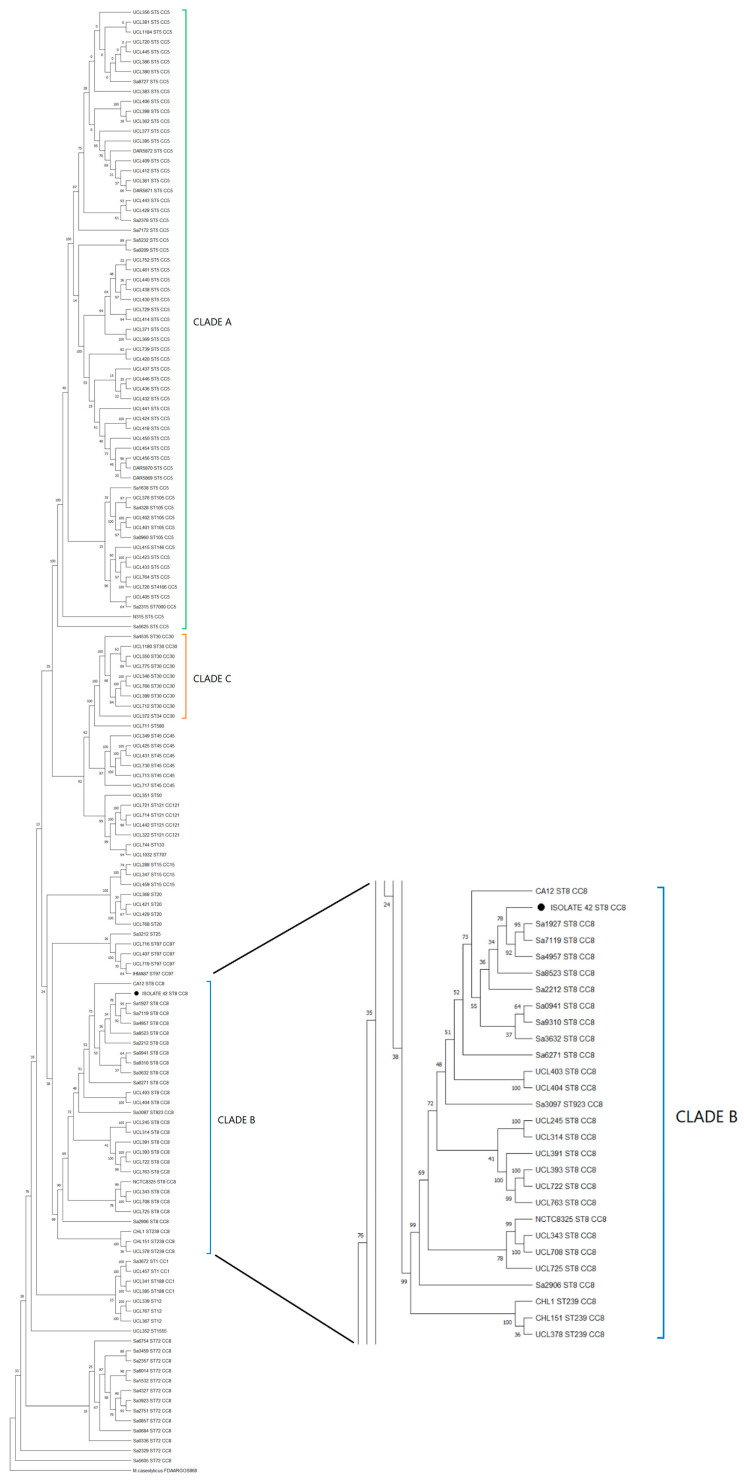
Maximum likelihood phylogenomic tree with bootstrap values (% of 200 replicates) of 145 Chilean MRSA strains. The three main clades (A, B, and C) highlighted in green, blue, and orange, respectively; isolate n°42 (● Tag) was grouped in Clade B. The tree was constructed from core gene alignment of 99% genes present in the genomes with 576,605 informative sites. Bootstrap consensus tree is shown.

**Figure 5 microorganisms-12-01284-f005:**
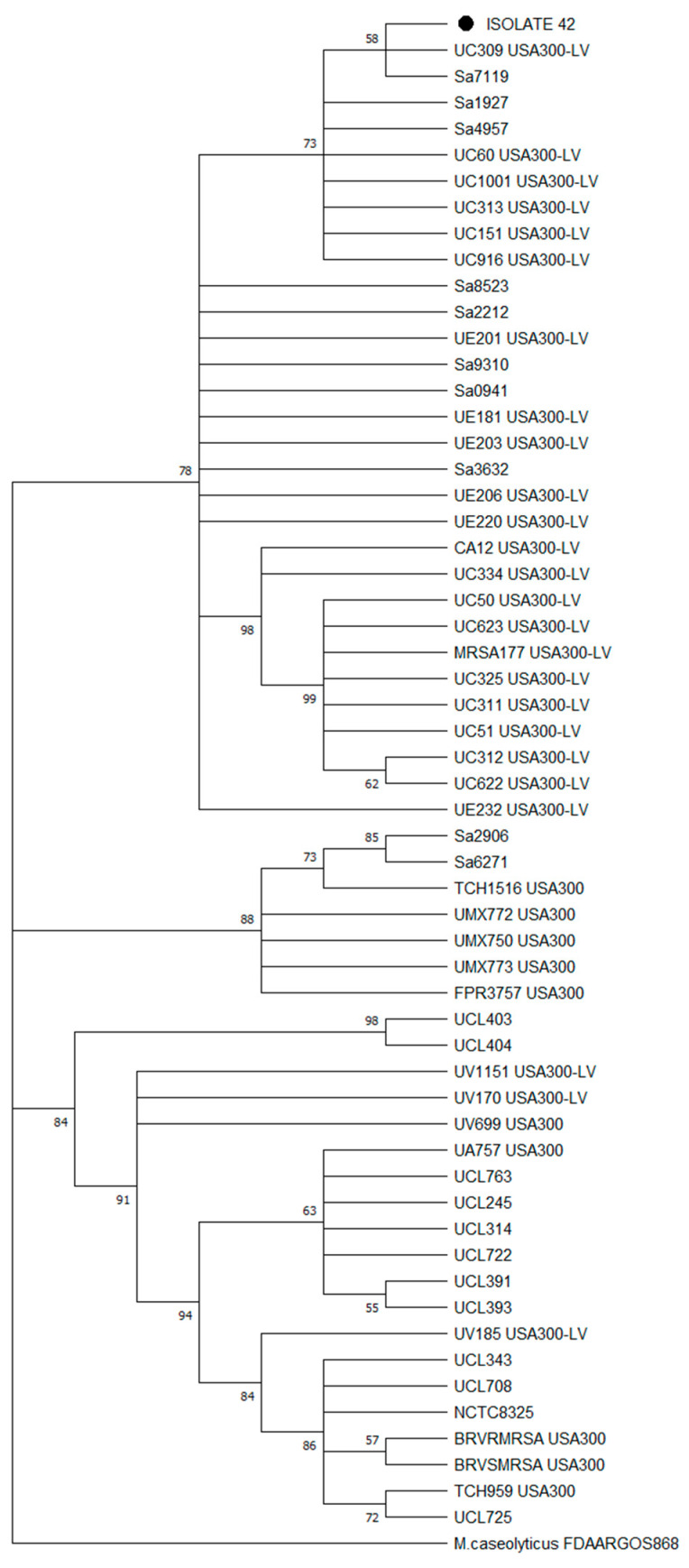
Maximum likelihood phylogenomic tree with bootstrap values (% of 1000 replicates) of 57 MRSA isolates, including isolate n°42 (● Tag), Chilean ST8 CC8 strains and Latin American USA300/USA300-LV strains identified by PFGE. The tree was constructed from a core gene alignment of 98% of genes present in the genomes with 1602 informative sites. Bootstrap consensus tree is shown.

**Table 1 microorganisms-12-01284-t001:** Antimicrobial susceptibility of MRSA isolates.

Antibiotic	MRSA *n* = 51 (%)		
	R	I	S
Oxacillin	48 (94.1)	-	3 (5.9)
Vancomycin	-	1(2.0)	50 (98.0)
Ciprofloxacin	43 (84.3)	2 (3.9)	6 (11.8)
Chloramphenicol	2 (3.9)	1 (2.0)	48 (94.1)
Gentamicin	25 (49.0)	-	26 (51.0)
Levofloxacin	44 (86.3)	1 (2.0)	6 (11.8)
Norfloxacin	45 (88.2)	1 (2.0)	5 (9.8)
Rifampin	6 (11.8)	-	45 (88.2)
Tetracycline	-	-	51 (100)

MIC and zone diameter breakpoint according to CLSI (30th ed.) for *S. aureus*: Oxacillin: S ≤ 2 µg/mL, R ≥ 4 µg/mL; Vancomycin: S ≤ 2 µg/mL, I 4–8 µg/mL, R ≥ 16 µg/mL; Ciprofloxacin: S ≥ 21 mm, I 16–20 mm, R ≤ 15 mm; Chloramphenicol: S ≥ 18 mm, I 13–17 mm, R ≤ 12 mm; Gentamicin: S ≥ 15 mm, I 13–14 mm, R ≤ 12 mm; Levofloxacin: S ≥ 19 mm, I 16–18 mm, R ≤ 15 mm; Norfloxacin: S ≥ 17 mm, I 13–16 mm, R ≤ 12 mm; Rifampicin: S ≥ 21 mm, I 16–20 mm, R ≤ 15 mm; Tetracycline: S ≥ 19 mm, I 15–18 mm, R ≤ 14 mm.

**Table 2 microorganisms-12-01284-t002:** Resistance profiles, virulence factors, and MLST of PVL-positive MRSA isolates.

Isolate No.	ST (CC)	Antibiotic Resistance Phenotype	Virulence Factors
11	5 (5)	OXA, GEN, CIP, LEV, NOR	*sdrC*, *sdrD*, *pvl*
12	923 (8)	OXA	*sdrC*, *sdrD*, *pvl*
13	5 (5)	OXA, GEN, CIP, NOR	*sdrC*, *sdrD*, *pvl*
14	72 (8)	OXA, CIP, LEV, NOR, RIF	*sdrD*, *pvl*
38	8 (8)	OXA, CHLO, CIP, LEV, NOR	*sdrD*, *pvl*
42	8 (8)	OXA, Vancomycin intermediate (VISA)	*sdrC, sdrD*, *pvl*
43	8 (8)	OXA	*sdrD*, *pvl*
44	5 (5)	OXA, GEN, CIP, LEV, NOR, RIF	*sdrD*, *pvl*
45	5 (5)	OXA, GEN, CIP, LEV, NOR, RIF	*sdrD*, *pvl*

CIP, ciprofloxacin; LEV, levofloxacin; NOR, norfloxacin; GEN, gentamicin; OXA, oxacillin; RIF, rifampin; CHLO, chloramphenicol; MLST, multilocus sequence typing; ST, sequence type; CC, clonal complex.

**Table 3 microorganisms-12-01284-t003:** Results of isolate n°42 sequence quality control.

ID Sequence	Phred 30 Average Quality	Number of Reads (Millions)	Reads Length (bp)
42_S34_R1_001	33.5	8.1	149
42_S34_R2_001	33.2	8.1	148
42_S34_trimmed_1P	34.7	6.3	129
42_S34_trimmed_2P	34.6	6.3	123

**Table 4 microorganisms-12-01284-t004:** Genome features of the assembled genome of isolate n°42.

Genome Feature	Result
Average coverage	290X
k-mer size used by Velvet assembler	111 bp
N50	453,788 bp
L50	3
Number of contigs	49
Genome size	2,800,637 bp
Number of protein-coding genes	2574
GC content	32%
Complete and single copy BUSCOs—Domain	100%
Complete and single copy BUSCOs—Phylum	99.5%
Complete and single copy BUSCOs—Class	99.7%
Complete and single copy BUSCOs—Order	100
23S rRNA genes	2
16S rRNA genes	3
5S rRNA genes	7
tRNA genes	62

**Table 5 microorganisms-12-01284-t005:** Virulence factor genes found in the assembled genome of isolate n°42.

Gene	Encoded Protein
*hlgA*	Gamma-hemolysin chain II precursor
*hlgB*	Precursor of component B gamma-hemolysin
*hlgC*	Gamma-hemolysin component C
*lukD*	Leukocidin D component
*lukE*	Leukocidin E component
*lukF-PV*	Panton Valentine Leukocidin F Component
*lukS-PV*	Panton Valentine Leukocidin S Component
*sek*	Enterotoxin K
*seq*	Enterotoxin Q
*aur*	Aureolysin
*splA*	Serine protease splA
*splB*	Serine protease splB
*splE*	Serine protease splE
*spa*	Staphylococcal protein A
*sdrC*	Serine-aspartate repeat proteins
*sdrD*	Serine-aspartate repeat proteins
*sdrE*	Serine-aspartate repeat proteins
*sasA*	*Staphylococcus aureus* surface protein A
*sasC*	*Staphylococcus aureus* surface protein C
*sasD*	*Staphylococcus aureus* surface protein D
*sasF*	*Staphylococcus aureus* surface protein F
*sasG*	*Staphylococcus aureus* surface protein G
*sasH*	*Staphylococcus aureus* surface protein H
*fnbA*	Fibronectin binding protein A
*fnbB*	Fibronectin binding protein B
*efb*	Extracellular fibrinogen-binding protein
*clfA*	Clumping factor A

**Table 6 microorganisms-12-01284-t006:** Antimicrobial resistance genes found in the assembled genome of isolate n°42.

Antimicrobial	Gene
Bacitracin	*bceA*
	*bceB*
	*bceR*
	*bceS*
Quinolones	*gyrA*
	*gyrB*
	*norB*
Tetracycline	*tetR*
	*tetA*
Vancomycin	*vraR*
	*vraS*
Fosfomycin	*fosB*
Bicozamycin	*bcr*
β-lactam	*mecA*

## Data Availability

The original contributions presented in the study are included in the article/Appendix A, further inquiries can be directed to the corresponding authors.

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
