# Peer review of "Molecular and Phylogenomic Analysis of a Vancomycin Intermediate Resistance USA300LV Strain in Chile"

_microorganisms, 2024, doi:10.3390/microorganisms12071284_

Round 1

Reviewer 1 Report

Comments and Suggestions for Authors

Introduction: Some Refs are missing

Materials and Methods: Methodological Biases exist, Refs are missing

(Possible) Limitations (?)

Please check attached file for more comments.

Author Response

We appreciate the comments made, which have been fully accepted since we believe they contribute to improving the article. Below is the detail of the changes made to the article:

  • Type of article: Research was added
  • In the abstract, in line 19, the sentences were written in a better way, MDR abbreviation was replaced by multi drug resistant in line 23; Line 26 "It" was added.
  • Line 51 reference added
  • Line 70, brackets removed
  • Line 71, reference was added
  • Line 96, reference was added
  • Line 116-121, Explanation to sample size was added, Inclusion/exclusion criteria to determine samples was added. 
  • Line 116, now 124 trademarks removed (Biomerieux)
  • Line 119, now 125 trademarks removed (Biomerieux)
  • Line 121, now 126 reference was added
  • Line 129, now 137 Reference and previous article as a base for antibiotic selection was added.
  • Line 129, now 139 reference was added
  • Line 129, now 136 trademarks removed (Santa Cruz Biotechnology, Inc., Dallas, TX, USA)
  • Line 134, Now 143 trademarks removed (HiMedia Laboratories, Mumbai, India)
  • Line 136, now 145 trademarks removed (Qiagen, Hilden, Germany)
  • Line 144, now 153 trademarks removed (Stratagene)
  • Line 159, now 167 reference was added
  • Line 436, now 445 some antibiotic resistances were added as examples
  • Line 500 now 509 references added
  • Limitation were added at the end of the conclusion (Lines 584-591)

Reviewer 2 Report

Comments and Suggestions for Authors

Dear authors,

The present paper is related with an important health risk and main topic nowatdays. Please, check the comments and observations describe into the text file. 

Author Response

We appreciate the comments made, which have been fully accepted since we believe they contribute to improving the article. Below is the detail of the changes made to the article:

  • Section 2.2 microorganism, symbol was changed µg
  • Section 2.23.1 microorganism, symbol was changed to µL
  • Section 2.3.2 MLST abbreviation was added
  • Section 2.4 Section reference  was corrected to 2.3.1
  • Table 1. S. aureus was corrected in italic font
  • Line 267, now 276 was added a space between table and paragraph
  • Supplementary table is a excel table annexed
  • Line 405, now 414, S. aureus was corrected in italic font
  • Line 500, now 509, S. aureus was corrected in italic font
  • Line 504, now 513, S. aureus was corrected in italic font
  • Line 538, now 546, S. aureus abbreviation was corrected
  • Line 568, now 579, S. aureus abbreviation was corrected

Reviewer 3 Report

Comments and Suggestions for Authors

Minor issues:

Lines 125-128: Please use the correct symbol for antibiotic disk quantity: µg., also use the µ symbol through the article.

Line 225 (Table 1 legend) please use also the  µ symbol.

Table 6: The mecA gene should'nt be present in the table? .

The focus of discution of vancomycin resistance is on vraRS genes ad is justified given the intermediate resistance phenotype, but the study does not address the potential presence or absence of vanA and vanB genes.

While the study effectively discusses molecular findings, there is less emphasis on the clinical implications of these findings for treatment strategies and public health policies.

Author Response

We appreciate the comments made to the article, which have been fully accepted. Below is the details of the change made

  • Lines 125-128: The symbol for antibiotic disk quantity: was corrected, the same dor the others µ symbol through the article.
  • Line 225 (Table 1 legend) the  µ symbol was corrected.
  • Table 6: The mecA gene was included, also in line 323 added "besides methicillin"
  • In lines 544, now line 554 is described that vanA and vanB genes were not found in the study.
  • Clinical implications were added at the end of the conclusion, Lines 582-584.
